# An Empirical Study of Promotion Pressure among University Teachers in China Using Event History Analysis

**DOI:** 10.3390/ijerph192215134

**Published:** 2022-11-16

**Authors:** Xiaoyan Liu, Lele Zhang, Haowen Ma, Haofeng Nan, Ran Liu

**Affiliations:** 1School of Languages and Communication Studies, Beijing Jiaotong University, Beijing 100044, China; 2Institute of Communication Studies, Communication University of China, Beijing 100024, China; 3School of Law, Guiyang University, Guiyang 550005, China; 4School of Public Health and Management, Wenzhou Medical University, Wenzhou 325035, China

**Keywords:** teacher promotion, promotion pressure, event history analysis, Kaplan–Meier, Cox regression

## Abstract

Objective: We sought to understand the status of promotion pressure among university teachers in China. This study explored the promotion duration and influencing factors among teachers in different disciplines of the social sciences. Methods: Using event history analysis, this study collected data regarding university teachers of China. The sample included 536 teachers who had been promoted from assistant to associate professor and 243 teachers promoted from associate to full professor. Our results revealed that the overall time required for promotion in the social sciences is relatively long. For those promoted from assistant to associate professor, the mean time for promotion was 14.155 years, with a median of 11 years, while for the transition from associate to full professor, the mean was 13.904 years with a median of nine years. Furthermore, in the survival function of the promotion duration, there is a stage pattern for both assistant to associate professor and associate to full professor. In addition, the Kaplan–Meier results showed that the mean promotion time in economics was the shortest. The Cox regression results indicated that males had a higher chance of promotion than females, and faculty members with doctoral degrees had a higher likelihood of promotion than those without. For those advancing from assistant to associate professor, the university of employment had significant positive effects on promotion. This paper provides empirical support for the current societal concerns regarding promotion pressure among university teachers.

## 1. Introduction

The stress mindset is the key antecedent of mental and physical well-being [1]. For university teachers in particular, stress and the resulting burnout in the current academic climate have a critical negative impact on well-being [2]. With recent reforms of the personnel promotion system, the dual pressures of teaching and research, and the lack of supporting job resources, Chinese university teachers face not only damage to their physical health but also an increase in their emotional exhaustion, which weakens their overall well-being [3,4,5]. Chinese university teachers are experiencing increasing pressure and health issues related to their teaching, research, and general living [6]. According to one survey, nearly 70% of Chinese university teachers reported greater work pressure, and more than 70% reported being in an unhealthy state, with some responses even noting premature death [7,8]. It is fair to say that the situation is becoming worse for young teachers. The damage to the quality of the mental health of Chinese university teachers is becoming even more serious than that affecting their physical quality of life [6]. Research has shown that university teachers are at a particularly high risk of job burnout, which directly affects their psychological health, potentially causing social and behavioral problems [9]. University teachers in China have long been impacted by occupational pressures due to the dual requirements of teaching and research. As of 2020, the total number of full-time university teachers in Chinese colleges was nearly 1.8 million [10]. The promotion pressure on young teachers is stronger and has gained widespread attention from both the media and academia [11]. With the introduction and implementation of the promotion system of the tenure track in China in 2014, the promotion pressure of university teachers has increased.

Searching the Web of Science on the topic of “faculty promotion”, we found that the number of studies on faculty promotion from different countries indicates a substantial increase in the last decade. However, details differ across different countries, and the chosen influencing factors have also differed across the various studies. For instance, studies from the U.S. usually focus on race, which apparently does not fit China’s particular context. Meanwhile, the data collected from different countries also have their own characteristics. For instance, Australian research combines gender differences in faculty promotion with discrimination, focusing on women’s status and gender differences [12]. Some Japanese research has paid more attention to the continuous work-related support of female faculty based on traditional Japanese gender role responsibilities [13], while other Japanese researchers have examined income differences in STEM academia according to gender, and they found that income differences between males and females still exist [14]. In the Chinese context, the gender factor has rarely been studied. Instead, Chinese researchers prefer to study teachers’ promotion policies and systems speculatively and qualitatively, yet only a few studies have focused on faculty promotion through quantitative research [15]. As for discipline preference, the data show that more studies have been conducted in the area of medicine and medical education [16].

Overall, the existing research has paid more attention to demographic factors such as race [17,18,19] and gender [12,20]. However, at present, our understanding of the differences in promotion rates according to different academic departments is lacking [21]. Furthermore, there are few comprehensive systematic studies on teachers’ promotion in disciplines of the social sciences as a whole in China, and the common characteristics of the different stages of promotion and the basic promotion rules affecting the 1.8 million Chinese full-time university teachers in these disciplines are still not clear enough.

In terms of methodology, the previous studies are largely based on traditional methods of multiple linear regression analysis. Goswami et al. (2022) used multivariable logistic regression analysis in a cross-sectional study to discover the effects of scholarly metrics [22]. Juraqulova et al. (2019) chose ordered probit models and found that a good performance in scholarly metrics will increases one’s hiring and/or promotional prospects in academia [23]. However, the study was not able to solve the problem of censoring and time-dependent covariates.

Concerning the above-mentioned research gaps and limitations, this study intended to use the event history analysis method to explore the promotion process of university teachers in order to provide support for, and improve the well-being of, university teachers.

## 2. Literature Review and Hypotheses

The factors influencing faculty promotion fall into three main categories: demographic factors, human capital factors, and school discipline factors.

### 2.1. Demographic Related Antecedents

Demographic factors include gender [20,24], race [18,25,26], and age [18,27].

Gender has drawn special attention in research. Studies have found that males have advantages over females regarding promotion probability, promotion duration, retention, academic rank, and income.

In terms of the promotion probability, in most disciplines, male teachers are more likely to be promoted than females. Kahn (2012) [12] used personnel records from 1999 to 2010 to analyze gender differences in teachers’ promotion and turnover at the University of New South Wales. The results showed that female associate professors were more likely to be promoted, while female assistants were less likely to be promoted or more likely to leave the university than men. Faria et al. (2013) [28] found that male teachers were promoted to associate professor at a rate of 13 percent more than females. Long et al. (1993) found that gender affects career promotion and that females are less successful than males in the promotion trajectories to associate and full professor [20].

As for the promotion duration, male teachers have been shown to have a shorter promotion time than female teachers, with research showing that male teachers are generally promoted from assistant to associate professor about five months earlier than female faculty members, and promotion from associate to full professor occurring about a year sooner for males compared to females [17]. Walker et al. (2020) conducted a descriptive statistical analysis using employment data from eight Aotearoa New Zealand universities and found that gender affected teachers’ promotion. Specifically, women were promoted later compared to men [29]. Messman et al. [30] found that the median time required for males to achieve their most recent promotion was 6.5 years, which was one year shorter than that of female teachers.

The duration difference between male and female university teachers varies across different disciplines. In an event history analysis of career promotion among science and engineering teachers at American universities, Kaminski and Geisler (2012) found that the median time of leaving an academic position was 10.9 years, and there was no significant difference in the rate of promotion between male and female faculty members. However, women faculty in mathematics were more likely to leave [31]. In the social sciences, the findings of decreasing gender differences in retention and resignation, as well as multi-stage title promotion, were corroborated [32].

In terms of academic rank, Zhu et al. (2021) found significantly more males among the academic surgery faculty than females [33]. The under-representation of female business faculty in senior academic positions is typical of Canadian higher education institutions [34].

In terms of retention and resignation, scholars’ views differ. Ponjuan et al. (2011) noted that gender matters in terms of satisfaction and attrition [35]. Meanwhile, Messman et al. [30] found that, compared to female faculty, the proportion of male faculty who had worked at the WSUSOM for less than 10 years was higher.

Income also differs greatly between females and males in regard to teachers’ promotion. Rao et al. (2018) analyzed gendered income differences and found that women were paid less compared to their male competitors [36]. Meanwhile, some scholars have suggested that there are more complex social factors at play behind the gender differences in academic careers. Wolfinger et al. [24] explored the effects of gender, fertility, and marital status on the career growth of academics and found that the difference in academic promotion between male and female faculty is not simply due to gender. Race and ethnicity are also important variables in career promotion [18,19]. Perna [25,26] applied descriptive statistics and logistic regression analysis to explore the effects of gender and ethnicity on the appointment of college university teachers. In terms of race, promotion rates for Hispanic and Black faculty were lower than those for White faculty at most academic medical centers [37].

Age is also a basic demographic factor. Wang and Sun (2013) found that the faculty promotion time for both the vice-senior title promotion period and senior title promotion period of regression are very significant [27]. These variables point to a pattern of decreasing probability of opportunity for promotion to full professor as one’s age increases [38].

Our initial goal in this study is to examine how gender affects the promotion of university teachers in China.

**H1.** 
*Gender has a significant influence on promotion duration, and there is a significant difference in promotion duration between male and female university teachers in China. Specifically, males have shorter promotion duration than their female peers at their various career stages.*


### 2.2. Human-Capital-Related Antecedents

Human Capital Factors refer to the personal knowledge, skills, and competencies ingrained in individuals that help to create individual well-being and social economy [39]. With regard to faculty promotion, this generally involves the academic level [25,26], graduating institution [27], school level [18], study abroad experience [40], post-doctoral experience [40,41], and in-service time [25,26].

Studies have explored whether the possession of a doctoral degree and the level of the graduating institution are among the important representative components of human capital [26]. Wang and Sun (2014) [27] found no effect of the academic background on the promotion process based on a case study of a university with a polytechnic background. Meanwhile, Li and Shen (2017) [42] conducted an empirical study of the overall college faculty and confirmed that human capital is an important factor which affects career development [42]. Zhang et al. (2019) focused on Changjiang scholars and found that their educational background affected their growth trajectories [40]. Susarla et al. [43] found that, for full-time academic oral and maxillofacial surgeons in the U.S., academic rank was significantly associated with the presence of a doctoral degree. Faria et al. [28] found that holding a Ph.D. earned before 2000 from a top U.S. university helped significantly with academic promotion.

The promotion duration at various types of schools can differ. Mallery et al. [18] divided institutions into five types, including historically Black colleges and universities, the Ivy League, minority serving institutions, and private and public institutions, and found that the institution type affects academic promotion assessment. While Ornstein et al. (2007) found that institutions differ statistically in all three disciplines during one’s promotion to full professor, the type of institution was not significant in the promotion step to associate professor [44]. Faria et al. (2013) found that it is more difficult to achieve academic promotion in research universities than in non-research institutions [28]. In China, the level of the university has a significant impact on the time required to achieve promotion, and the higher the prestige of the university is, the shorter one’s average promotion time, in the case of university teachers, will be [42].

Concerning data availability, the following hypotheses aimed to examine how the possession of a doctoral degree, level of one’s graduating university, and level of employment at the university affect the promotion duration of university teachers in China:

**H2.** 
*Having a doctoral degree significantly influences the promotion duration, and there is a significant difference between the promotion duration of university teachers with or without a doctoral degree. Specifically, university teachers with a doctoral degree have a shorter promotion duration than those without at the various career stages.*


**H3.** 
*The level of the graduating university has a significant influence on the promotion duration, and there is a significant difference between university teachers who graduated from various levels of institutions. Specifically, university teachers who graduated from a higher-level institution have a shorter promotion duration than those who graduated from a lower-level institution at all the various career stages.*


**H4.** 
*One’s level of employment at a university has a significant influence on the promotion duration, and there is a significant difference between and among the different levels of employment at universities at all the various career stages.*


### 2.3. Discipline-Related Antecedents

Discipline factors refer to the department category [21,44].

Discipline matters in terms of teachers’ promotion. A study of Canadian university teachers found that teachers of science and engineering were promoted more quickly and that their affiliation and discipline had greater impacts on their career development than their gender [42]. In exploring the relationship between gender, race, discipline, and promotion outcomes, Durodoye et al. (2020) found that faculty promotion rates varied by discipline and that discipline differences had a greater impact on promotion rates than gender or racial differences [45]. In an analysis of faculty biographies in both engineering and immunology and management science, the institution had a significant impact on one’s career promotion in both disciplines [41]. A comparison based on the disciplines of mathematics, history, and education also found that the discipline type led to significant differences in the promotion from assistant to associate professor, while there was no significant difference found in the promotion from associate to full professor [21]. Xierali et al. (2021) found that the discipline and education level affected the promotion of the faculty of medical schools [46]. Yue (2020) analyzed the probability of promotion and its influencing factors in the disciplines of history, education, and mathematics at colleges located in East China [21]. Other scholars have analyzed the gender, race, and career growth of faculty members across disciplines at four universities and found large differences in tenure probability rates between disciplines [45].

However, our understanding of the differences in promotion rates according to the academic department is currently lacking [21], with few comprehensive, systematic studies on faculty promotion within the disciplines of the social sciences as a whole in China. Our fifth hypothesis set up a comparison between and among the different disciplines of the social sciences, as well as differences between different universities of employment:

**H5.** 
*Discipline has a significant influence on promotion duration, and there are significant differences between and among university teachers in different disciplines.*


## 3. Methods

### 3.1. Event History Analysis

Event history analysis is a statistical method used to analyze the occurrence and timing of events within a given time, which does enable some cases to be omitted [36]. Event history analysis deals with data obtained by observing individuals over time. Thus, it focuses on the events which occur in the lives of the individuals [47]. It has different terminologies in different disciplines, such as event history analysis (sociology), reliability analysis (engineering), failure time analysis (engineering), duration analysis (economics), and transition analysis (economics). These different terminologies do not reflect any real differences in technique [48]. In this study, we chose the name event history analysis, which we took as being the most appropriate.

Event history analysis can explain why certain individuals have a higher risk of experiencing the event(s) of interest [49]. It can be used to solve two problems that traditional multivariate statistical methods cannot solve, including censoring and time-dependent covariates [19]. A hazard model is able to handle censored observations containing partial information and covariates that change dynamically during the observing period. These two distinguishing features differentiate it from other regression models [48,49].

### 3.2. Definition of Key Concepts

An event is defined as a transition from one stage to another [47]. In the current study, the event of interest was defined as successful promotion to the next rank and title, specifically the promotion from assistant to associate professor or from associate to full professor.

The promotion duration (i.e., survival time) refers to the observed survival time of a promotion. In this study, the time of promotion is the time period of interest and usually refers to how many years it takes for university teachers to be promoted to the next higher rank, specifically the years required to move from assistant to associate professor or from associate to full professor.

Censoring occurs when incomplete information is available about the survival time of only some individuals [50]. That is to say, censoring occurs when we have some information about the individual survival time without knowing some individuals’ survival time. Censored data constitute a key characteristic that distinguishes event history analysis from other areas in statistics [50], such as those based on the most basic distinction of left or right censoring [19].

### 3.3. The Kaplan–Meier Method

The Kaplan–Meier (KM) method, which is a nonparametric maximum likelihood estimator, was put forward by Kaplan and Meier (1958) to handle incomplete observations [51].

The survival function, $\hat{S}\left(t\right)$, is defined as the probability of surviving in a given length of time. Suppose there are *k* distinct event times, *t*_1_ < *t*_2_ < … < *t_k_*. At each time *t_j_*, there are *n_j_* individuals who are said to be at risk of an event. At risk means that they have not experienced the event, nor have they been censored from it prior to time *t_j_*. If any cases are censored at exactly *t_j_*, they are also considered to be at risk at *t_j_*. Let *d_j_* be the number of individuals who are promoted at time *t_j_* [48].

The KM estimator, then, is defined as:(1)$$\hat{S}\left(t\right)=\underset{j:t_{j}\leq t}{\prod }\left[1-\frac{d_{j}}{n_{j}}\right]$$

In this study, $\hat{S}\left(t\right)$ is defined as the probability of university teachers not being promoted or continuing in the same title rank throughout our observation time.

### 3.4. The Cox Proportional Hazards Model

In 1972, the British statistician David Cox developed the proportional hazards model, which derives robust estimates of covariate effects using a proportional hazards assumption [48,52]. In this study, this model allowed us to estimate the hazard (or risk) of promotion of university teachers given their independent variables.

This model can be written as follows:(2)$$h_{i}\left(t\right)=\lambda_{0}\left(t\right)\exp\left\{\beta_{1}x_{i1}+\ldots +\beta_{k}x_{ik}\right\}$$

In this study, *X*_1_, *X*_2_, … *X_k_* are risk factors, which are the related factors that affect the promotion duration (survival time). *β*_1_, *β*_2_, … *β_k_* are regression coefficients. *h*_0_(*t*) is the baseline hazard and represents the hazard in which all of the independent variables are equal to zero. If *β*_1_ is greater than 0, this indicates that the covariate is a risk factor [48].

The function *λ*_0_(*t*) can be regarded as the hazard function for university teachers whose covariates all have values of 0. Taking the logarithm of both sides, we can rewrite the model as:(3)$$\log h_{i}\left(t\right)=\alpha \left(t\right)+\beta_{1}x_{i1}+\ldots +\beta_{k}x_{ik}$$

Taking the ratio of the hazards for two faculties *i* and *j*, we can apply them to the equation thusly (3):(4)$$\frac{h_{i}\left(t\right)}{h_{j}\left(t\right)}=\exp\left\{\beta_{1}\left(x_{i1}-x_{j1}\right)+\cdots +\beta_{k}\left(x_{ik}-x_{jk}\right)\right\}$$

### 3.5. Data

Data were collected from university teachers working in China in one of four different social science disciplines, including journalism and communication, sociology, economics, and law, through the official websites of the universities, as well as other websites which further supplemented our information.

The Catalogue of Undergraduate Specialties of General Higher Education Institutions of the Ministry of Education of China guided our selection of the universities to be included in the study, which yielded a list of 155 colleges and universities in mainland China with the above four disciplines as undergraduate study options.

China uses a grading system for higher education, and the key national universities are divided into categories which include Project 985 and Project 211, as well as other ordinary rankings. Project 985 is a national governmental project aiming to create world-class universities during the 21st century. On 4 May 1998, President Jiang Zemin declared that “China must have a number of first-rate universities of an international advanced level”, launching Project 985. Project 211, meanwhile, is the Chinese government’s newest endeavor aimed at strengthening roughly 100 higher education institutions and key disciplinary areas as a national priority in the 21st century. There are 112 universities included in Project 211.

Thus, of the 155 universities included in the Catalogue of Undergraduate Specialties, 22 are a part of Project 985, 35 area part of Project 211, and 98 are ordinary schools. A stratified random sampling was then conducted for the three types of universities, resulting in 2 Project 985 colleges and universities, 4 Project 211 academic institutions, and 10 ordinary college or universities. Out of these universities, after missing data were deleted, 536 faculty survival data were obtained for individuals promoted from assistant to associate professor, and 243 faculty survival data were obtained for those promoted from associate to full professor.

In the end, the number of journalism and communication teachers included was 111, the number of teachers of sociology was 105, the number of faculty of economics was 180, and the number of faculty of law was 259 (See Table 1).

## 4. Results

### 4.1. KM Analysis of Promotion Duration

For those promoted from assistant to associate professor, the overall mean of the promotion duration was 14.155 years, and the overall median was 11 years. As of the time of our data collection, assistant professors in law spent the longest time working to attain a promotion, with a mean of 16.112 years.

For the promotion from associate to full professor, the overall mean duration was 13.904 years, and the median was 9 years. For this stage of promotion, the mean time was the longest among associate professors in journalism and communication, at 18.722 years.

For both promotion stages, university teachers of economics had the shortest time before receiving a promotion, with a mean of 9.439 years for the first stage from assistant to associate professor and a mean of 10.886 years for associate to full professor (See Table 2).

### 4.2. Univariate Analysis of Promotion Duration

For the promotion stage from assistant to associate professor, at the 99% confidence level, the log rank test results showed that there were significant differences in promotion duration between the various disciplines measured (*p* = 0.000 < 0.001). Other independent variables, including gender (*p* = 0.000 < 0.001), education level (*p* = 0.000 < 0.001), university of employment (*p* = 0.000 < 0.001), and university of graduation (*p* = 0.000 < 0.001) also had significant impacts on the promotion duration (See Table 3).

For the promotion stage from associate to full professor, at the 99% confidence level, the log rank test results showed that gender (*p* = 0.000 < 0.001) and the education level (*p* = 0.000 < 0.001) had significant impacts on the duration of the promotion. Meanwhile, at the 95% confidence level, the other three independent variables (i.e., discipline, university of employment, university of graduation) were not found to be significant in terms of the duration of the promotion. Thus, all the independent variables were applied to a Cox model for further analysis.

### 4.3. Cox Model Results Analysis

#### 4.3.1. Stages of the Duration of the Promotion

Our results showed that 11 years appeared to be a turning point for faculty promotion from assistant to associate professor. For the period of 0 to 11 years, the curve of the cumulative survival is steep and drops rapidly, while after the 11-year point, the curve slows down and becomes relatively smooth (see Figure 1).

For the stage from associate to full professor, the duration appeared to be divided into three stages. In the first stage (0 to 5 years), the curve descends quickly; in the second stage (6 to 10 years), the curve becomes relatively steady; and in the final stage (over 10 years), almost no change is seen (see Figure 2).

The results of the Omnibus tests of the model coefficients for the assistant to associate professor promotion stage were −2 log likelihood = 2407.427 and *p* = 0.000. The results of the Omnibus test of the model coefficients for the stage from associate to full professor were −2 log likelihood = 828.125 and *p* = 0.000.

#### 4.3.2. Effect of Disciplines

With the other variables controlled, at the 95% confidence level, no significant effects of the disciplines on the promotion duration from assistant to associate professor (*p* = 0.110 > 0.05) or from associate to full professor (*p* = 0.633 > 0.05; see Table 4, Figure 3, Table 5, and Figure 4) were found.

In the exploration of the influencing factor of the discipline, although it was not found to be significant, the results are still meaningful in the case of our data.

#### 4.3.3. Effect of Gender

For the promotion stage of assistant to associate professor, gender had a significant effect on the promotion duration (*p* = 0.002 < 0.01) (see Table 4 and Figure 5). Compared with males, females’ likelihood of promotion decreased by 36.1% (i.e., 1.000–0.639%), meaning that males found it easier to gain promotions and were more likely to attain a promotion than their female peers. On average, females spent 2.59 years more than males as assistant professors before being promoted to associate professor and took 8.367 years longer to be promoted from associate to full professor (see Table 6).

Gender also had a significant effect on the promotion duration for the stage from associate to full professor (*p* = 0.000 < 0.01; see Table 5 and Figure 6). Females’ likelihood of promotion decreased by 69.4% (i.e., 1.000–0.306%) compared to their male colleagues, indicating that it was more difficult for females to receive promotions compared to their male colleagues under the same conditions.

The slower rate of promotion among female university teachers in the Chinese context could be due to gender-related roles in the family. For instance, female university teachers might be less likely to be promoted because of childbirth [53], or it could be due to gender-biased perceptions in society, such as social expectations and the perception of gender stereotypes and biological constraints [54]. Additionally, this would surely reduce the well-being of female teachers more than that of their male peers. Thus, optimized and loose policies need to be implemented to better support female faculty.

#### 4.3.4. Effect of the Doctoral Degree

For the stage from assistant to associate professor, when compared to university teachers with a doctoral degree, university teachers without a doctoral degree reported a significant difference in their promotion time (*p* = 0.000 < 0.001; see Table 4). The likelihood of the promotion of university teachers without a doctoral degree was decreased by 76% (i.e., 1.00–0.24%), which means that it took more time for them to be given a promotion.

For the stage from associate to full professor, there was a significant difference in the duration of the promotion when comparing university teachers with and without a doctoral degree (*p* = 0.03 < 0.05; see Table 5). The likelihood of the promotion of university teachers without a doctoral degree was decreased by 64.2% (i.e., 1.00–35.8%) compared with those with a doctoral degree.

#### 4.3.5. Effect of the University of Employment

For the stage from assistant to associate professor, the university of employment was shown to have a significant effect on the promotion duration (*p* = 0.000 < 0.001; see Table 4). University teachers working in Project 985 universities tended to have the shortest time before being promoted. This shows that there was a significant difference in the promotion duration between university teachers from Project 985 and ordinary universities. Compared to university teachers from Project 985 universities, the likelihood of promotion of university teachers working in ordinary universities was decreased by 56.9% (i.e., 1.000–0.431%). Although a significant difference was not found in the promotion duration experienced between faculty from Project 985 and those from Project 211 universities, we nonetheless found that the likelihood of the promotion of university teachers working in Project 211 universities was decreased by 20.3% (i.e., 1.000–0.797%).

According to funding statistics from the Ministry of Education, 20% of universities obtain 95% of the ministry’s research funding, while 5% of the funding is allocated to the remaining 80% of the universities. The management of university teacher career promotions should take care to avoid the Matthew effect of resource allocation between different types of schools and focus on the academic career development of teachers with diverse talents in all schools, regardless of the type.

Meanwhile, for the promotion stage of associate to full professor, the university of employment did not seem to have any effect on the duration (*p* = 0.730 > 0.05; see Table 5).

The university of graduation was not found to be significant in terms of the promotion duration in either stage.

## 5. Discussion

### 5.1. Theoretical and Practical Implications

Promotion pressure is critical to the well-being of university teachers. A longer promotion duration results in uncertainty, stress, and burnout and has negative impacts on university teachers’ well-being [2]. Using event history analysis, this study examined the survival distribution of the promotion duration of university teachers from various universities in China and its influencing factors. This study contributes to the literature on the academic promotion and well-being of university teachers in several aspects.

Firstly, this study expands the scope of our understanding of university teachers’ promotion, which, until now, has focused mainly on Western contexts and natural sciences, such as medicine and health. The current study, instead, focused on China and the social sciences, considering journalism, sociology, economics, and law. Furthermore, studies comparing different disciplines are scarce. Thus, the data in this study provide a meaningful supplement to the existing data, which come primarily from Western countries.

Secondly, the current study proposes a more complete and systematic analytical framework for the influences on university teachers’ promotion, including demography, human capital, and discipline characteristics.

Thirdly, our research provides a more fine-grained conclusion and further analyzes the promotion stage patterns which have been neglected in the previous literature. As shown in our study, the transition from assistant to associate professor can be divided into two stages, shifting at the 11-year point. From 0 to 11 years, the curve of the cumulative survival is steep and drops rapidly, while after the 11-year point, the curve tends to slow down and becomes relatively smooth. Meanwhile, there are three stages in the transition from associate to full professor. During the first stage (0 to 5 years), the curve descends rapidly; in the second stage (6 to 10 years), the curve becomes steady; and in the final stage (more than 10 years), there is almost no change, which indicates the difficulty for remaining faculty who have been associates for over 10 years, whose progress becomes slow, with less opportunity for promotion.

Fourthly, our study contributes to methodological developments by overcoming the shortcomings of traditional linear regression, utilizing the relatively newly introduced event history analysis method to handle the problem of censoring and time-dependent covariates.

Our findings inform the consideration and discussion of the ways through which we may further optimize the current career promotion system of university teachers and improve the career development opportunities of university teachers so as to create a more comprehensive and flexible academic environment to ensure that the faculty teaching, research, and career load university teachers of all disciplines can provide a healthy environment of competition.

Our research also has several practical implications for the improvement of the well-being of university teachers through reducing promotion pressures on university teachers in China. Firstly, our findings demonstrated that there are two different stage patterns of promotion from assistant to associate professor and associate to full professor. With this knowledge, institutional personnel departments should consider the various care policies that they can provide to faculty at different promotional stages. For teachers in their first promotional stage, institutions should reduce the assessment standards appropriately for both research and teaching or allow teachers to choose to focus on specific research or teaching according to their own interests and abilities, as well as the school’s situation. Secondly, the current study found that gender disparities continue to exist in terms of promotion pressure on university teachers in the social science disciplines in China. This most certainly reduces the well-being of the female teachers even more than that of their male peers. Thus, optimized and flexible policies should be implemented to better support female faculty. Thirdly, different disciplines also have different promotion pressures. In contrast to economics, the mean promotion times of teachers in journalism and communication, sociology, and law were all much longer. Therefore, educational authorities and institutional personnel departments must create different performance appraisal policies for teachers in different disciplines. More importantly, more attention should be given to faculty in these three specific disciplines over those in economics. Performance appraisals should not be carried out blindly according to titles, and faculty should be offered more union benefits, and holistic encouragement should be provided to improve their overall well-being.

### 5.2. Limitations and Future Research

The current study does have limitations. In terms of the data, there are still some problems regarding incomplete information due to differences in the data disclosure, as well as the belated updates of some of the college and university websites, which led to missing or inaccurate data. Future research should work to further clarify the data and details of the existing discipline samples, as well as expand the sample size by including, for example, faculty of political science, management science, and other social sciences to build a more comprehensive analysis. Meanwhile, due to our research framework, other variables that may also affect faculty promotion were not taken into consideration, such as the published papers and funding levels, which should also be considered in future studies.

## 6. Conclusions

Using event history analysis, this study explored the academic promotion time, as well as its influencing factors, among university faculty from a variety of social science disciplines in China. The research findings showed that the overall time for promotion in the social sciences is relatively long, with different patterns evident for each of the two different promotion stages, both of which are certain to increase the promotional pressure and reduce the well-being of university teachers. As for the influencing factors of gender, the doctoral degree, and level of employment at the university, all were shown to have significant effects on the promotion duration of university teachers. This paper provides empirical support for the current societal concerns regarding the promotion pressure on university teachers.

## Figures and Tables

**Figure 1 ijerph-19-15134-f001:**
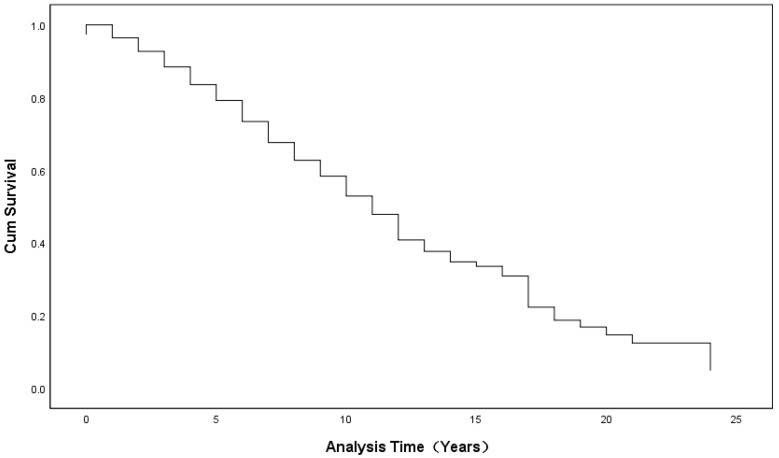
Survival Functions for the Mean of the Covariates (Assistant to Associate).

**Figure 2 ijerph-19-15134-f002:**
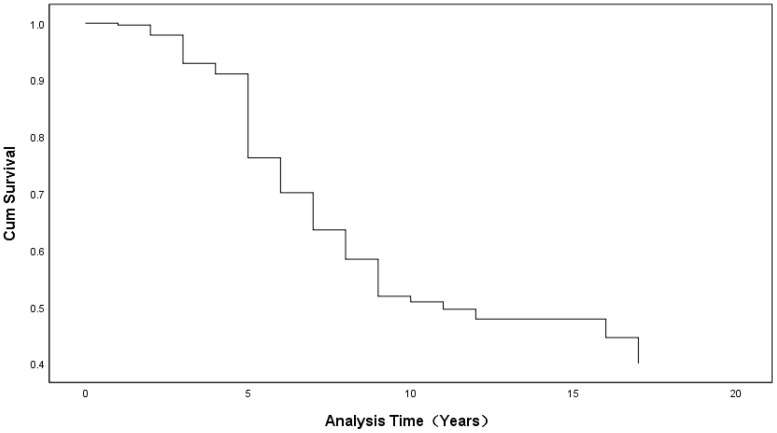
Survival Functions for the Mean of the Covariates (Associate to Full Professor).

**Figure 3 ijerph-19-15134-f003:**
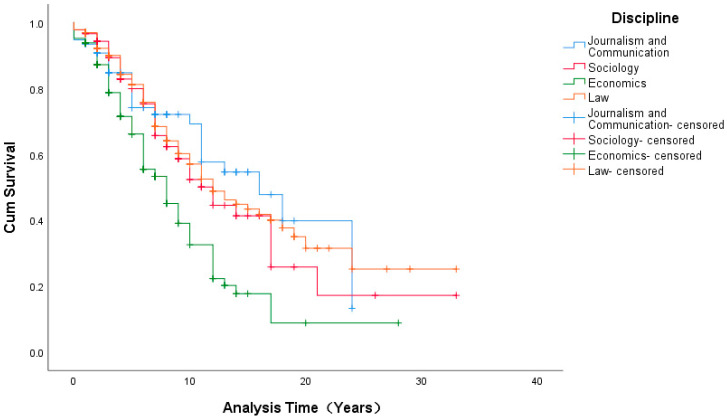
Survival Functions of the Mean of the Covariates from Assistant to Associate (Disciplines).

**Figure 4 ijerph-19-15134-f004:**
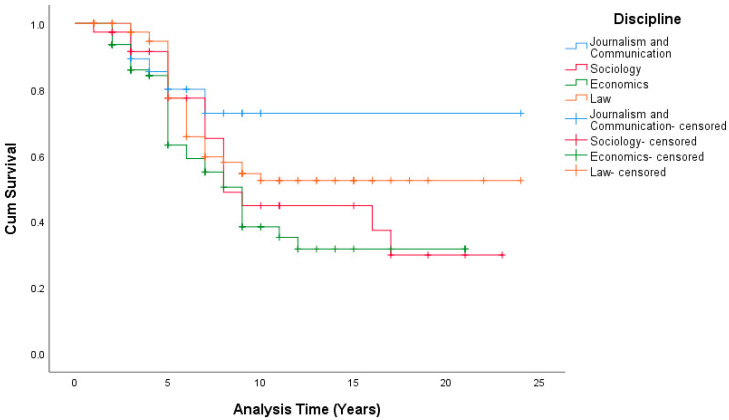
Survival Functions From Associate to Full Professor (Disciplines).

**Figure 5 ijerph-19-15134-f005:**
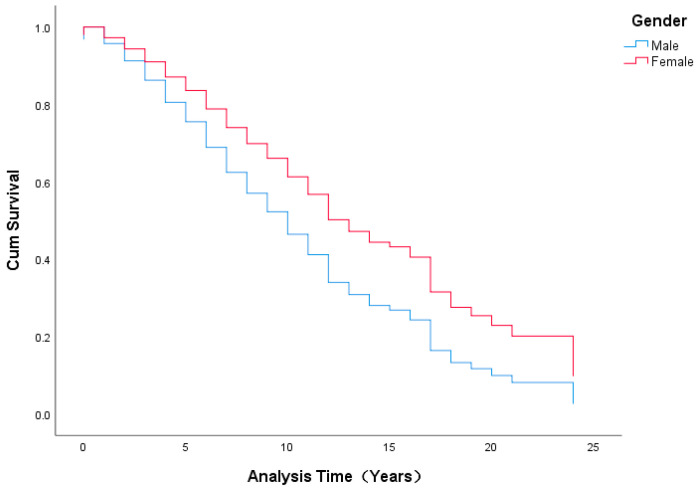
Survival Function for Gender in Promotion From Assistant to Associate Professor.

**Figure 6 ijerph-19-15134-f006:**
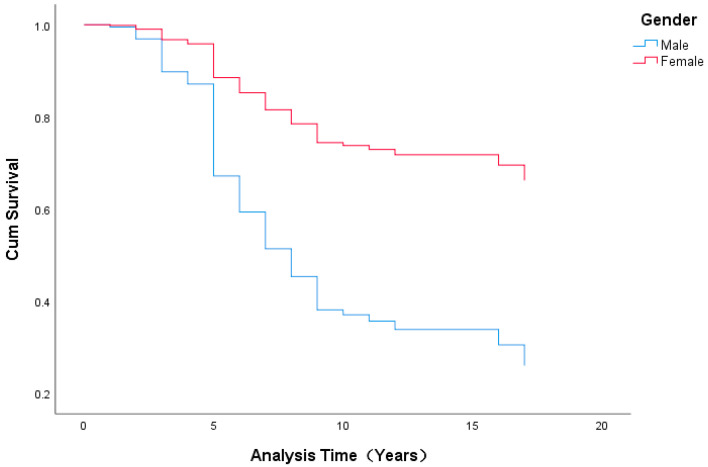
Survival Function for Gender in Promotion from Associate to Full Professor.

**Table 1 ijerph-19-15134-t001:** Proportion of Each Discipline.

	Assistant to Associate Professor				Associate to Full Professor			
			Censored				Censored	
Disciplines	Total	Events	N	Percent	Total	Events	N	Percent
Journalism and communication	76	28	48	63.2%	35	6	29	82.9%
Sociology	88	40	48	54.5%	38	17	21	55.3%
Economics	145	81	64	44.1%	83	35	48	57.8%
Law	227	83	144	63.4%	87	32	55	63.2%
Total	536	232	304	56.7%	243	90	153	63.0%

**Table 2 ijerph-19-15134-t002:** Means and Medians of Promotion Duration of Different Disciplines.

		Assistant to Associate Professor				Associate to full professor			
		95% Confidence Interval				95% Confidence Interval			
	Discipline	Estimate	SE	Lower	Upper	Estimate	SE	Lower	Upper
Mean	Journalism and communication	14.964	1.325	12.366	17.561	18.722	1.923	14.952	22.492
	Sociology	14.284	1.681	10.989	17.578	12.767	1.540	9.749	15.786
	Economics	9.439	0.894	7.687	11.191	10.886	1.045	8.838	12.934
	Law	16.112	1.232	13.697	18.527	15.412	1.117	13.223	17.601
	Total	14.155	0.734	12.716	15.594	13.904	0.744	12.447	15.362
Median	Journalism and communication	16.000	3.613	8.919	23.081				
	Sociology	12.000	1.462	9.134	14.866	8.000	0.964	6.110	9.890
	Economics	8.000	0.916	6.204	9.796	9.000	0.851	7.332	10.668
	Law	12.000	1.535	8.991	15.009				
	Total	11.000	0.617	9.791	12.209	9.000	2.790	3.532	14.468

**Table 3 ijerph-19-15134-t003:** Univariate Analysis of the Duration of the Promotion.

	Assistant to Associate Professor						Associate to Full Professor					
	Log Rank			Breslow			Log Rank			Breslow		
Disciplines	χ^2^	df	Sig.	χ^2^	df	Sig.	χ^2^	df	Sig.	χ^2^	df	Sig.
Gender	16.912	1	0.000	20.570	1	0.000	27.446	1	0.000	25.576	1	0.000
Education level	61.175	1	0.000	39.401	1	0.000	12.441	1	0.000	8.493	1	0.004
Discipline	23.602	3	0.000	16.834	3	0.001	6.360	3	0.095	5.840	3	0.120
University of employment	47.258	2	0.000	30.469	2	0.000	2.798	2	0.247	2.211	2	0.331
Graduating university	19.595	4	0.001	11.745	4	0.019	5.778	4	0.216	4.282	4	0.369

Note: graduate universities refer to those which offer the highest degree possible for university teachers.

**Table 4 ijerph-19-15134-t004:** Cox Model Results (Assistant to Associate Professor).

	B	SE	Wald	df	Sig.	Exp(B)
University of employment			23.784	2	0.000	
Project 985 (=0)						
Project 211	−0.227	0.172	1.739	1	0.187	0.797
Ordinary university	−0.842	0.179	22.032	1	0.000	0.431
Gender	−0.449	0.148	9.153	1	0.002	0.639
Discipline			6.026	3	0.110	
Journalism and communication (=0)						
Sociology	−0.154	0.258	0.355	1	0.551	0.858
Economics	0.320	0.234	1.877	1	0.171	1.378
Law	0.074	0.224	0.110	1	0.740	1.077
University of graduation			6.646	4	.156	
Project 985 (=0)						
Project 211	−0.273	0.171	2.535	1	0.111	0.761
Overseas college	−0.332	0.203	2.668	1	0.102	0.718
Ordinary university	−0.124	0.320	0.150	1	0.699	0.883
Chinese Academy of Sciences System	−0.894	0.512	3.045	1	0.081	0.409
Degree	−1.425	0.255	31.263	1	0.000	0.240

**Table 5 ijerph-19-15134-t005:** Cox Model Results (Associate to Full Professor).

	B	SE	Wald	df	Sig.	Exp(B)
University of employment			0.630	2	0.730	
Project 985 (=0)						
Project 211	−0.161	0.282	0.327	1	0.568	0.851
Ordinary university	−0.223	0.294	0.576	1	0.448	0.800
Gender	−1.184	0.311	14.439	1	0.000	0.306
Discipline			1.716	3	0.633	
Journalism and communication (=0)						
Sociology	0.187	0.493	0.143	1	0.705	1.205
Economics	0.437	0.458	0.910	1	0.340	1.548
Law	0.161	0.458	0.123	1	0.725	1.175
University of graduation			0.592	4	0.964	
Project 985 (=0)						
Project 211	−0.211	0.288	0.539	1	0.463	0.809
Overseas college	0.003	0.330	0.000	1	0.992	1.003
Ordinary university	0.016	0.579	0.001	1	0.977	1.017
Chinese Academy of Sciences System	0.098	0.762	0.017	1	0.898	1.103
Degree	−1.026	0.473	4.714	1	0.030	0.358

**Table 6 ijerph-19-15134-t006:** Means and Mediansof Promotion Duration of the Genders.

		Assistant to Associate Professor				Associate to Full Professor			
		95%ConfidenceInterval				95%ConfidenceInterval			
	Discipline	Estimate	SE	Lower	Upper	Estimate	SE	Lower	Upper
Means	Male	12.677	0.895	10.923	14.431	11.110	0.862	9.420	12.800
	Female	15.267	0.957	13.392	17.142	19.477	1.088	17.345	21.608
	Total	14.155	0.734	12.716	15.594	13.904	0.744	12.447	15.362
Medians	Male	9.000	0.735	7.559	10.441	7.000	0.658	5.711	8.289
	Female	14.000	1.950	10.179	17.821				
	Total	11.000	0.617	9.791	12.209	9.000	2.790	3.532	14.468

## Data Availability

The raw data supporting the conclusions of this article will be made available by the authors without undue reservation.

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
