# Peer review of "An Empirical Study of Promotion Pressure among University Teachers in China Using Event History Analysis"

_ijerph, 2022, doi:10.3390/ijerph192215134_

Round 1
Reviewer 1 Report
Dear authors, the research presented in the paper is interesting, but it needs further development:
1. check English spelling, because there are a number of misspellings that need to be adjusted
2. check English meaning, because some of the formulations make little sense or the meaning is hard to understand.
3. check the cross-reference, because there is this message in the first part of the article: Error! Reference source not found.]
4. consider expanding the Research Methodology section, to make it even more clear
5. consider adding chapters such as: Discussions, Limitations, Future research
Reviewer 2 Report
This paper analyzes the time needed for a Chinese faculty member in order to get promotion within the China Academic career. The article is not particularly novel and it presents unsursprising conclusions. Moreover, it needs an extensive English but also general editing. Nonetheless, the authors are able to link interestingly their results with some relevant conclusions. For this reason, I think that after some work for improving the overall quality of the article, it could be taken into consideration for a publication. Please, find below my detailed comments.
1) I found at least 8 missing references. Are them self-citation? This journal does not blind authors' names, so please uncover these references for a further control of relevance;
line 52-54) Does this paragraph refer to China or to the whole academic world?
2) You should create an introduction and a literature review, separately. The introduction should cover a clear statement of the issue under analysis and broadly its relevance. Moreover, in the introduction you need to desribe sortly your analysis, the context on which the analysis is framed, your resarch questions, some hints about your results, and the novelties you are adding with your study. Literature review has to include the state of the art: as already describe in your actual introduction is sufficiently good.
line 167-168) You should explain what censoring is, or taking it as given. Your sentence in your actual formulation is useless.
Equation1) Equation 1 it is poorly explained, S and s are never defined and probably you're missing a LaTeX package so that the equation layout is messed up.
Line 176-182) This paragraph is so confused. What is Inh(t)? What is the online influence? And the risk rate of deaths? Once you have clarified your survival framework, you should abandon the traditional medical terminology.
line 228) Why in this case the Median is not a round number? How did you calculate it?
Line 245) Are you sure that the turning point is 15 years? It seems to me 11 or 12.
Line 276) You state that gender has a significant effect on risk but in Table 4 statistics are missing. What's going on?
Table 5) How do you obtain p-values for baseline factors (e.g. Project985 or J&C)?
4) Figure 7 and 8 are confused and never mentioned in the text. What's the difference between censored and non-censored?
Reviewer 3 Report
This is an interesting paper but there are some issues that must be addressed before it can be published:
Abstract
1. Instead of mechanically describing statistical results, such as "Cox regression results indicated that males had a higher risk of promotion than females", it is difficult for readers to understand whether men are more likely or less likely to be promoted.
Introduction
1. There are a lot of "[Error! Reference source not found.]. This is due to a problem with the author's reference software, but the author should completely correct these problems when revising.
2. The logical confusion of the literature review. Since the authors found by searching the literature that "Influencing factors of faculty promotion can fall into three main categories, demo-graphic factors, human capital factors and school discipline factors." So why doesn't the author directly divide the literature review into three parts in the process of writing? I didn't see any literature on age in the literature review.
3. The author needs to sort out the literature on teacher promotion in different countries, and the national conditions of different countries are very different, and they are directly and confusingly put together, which is easy to cause readers to misunderstand.
4. Don't use "Research Question" and formulate your research hypothesis based on the purpose of the research.
Methods
1. I don't understand the J&C and SOC in Table and should not use abbreviations arbitrarily if there is no specific purpose.
Other questions
1. Please write Discussion, Conclusion, and Limitation separately
2. The Discussion section should focus on how your research differs from existing research
3. The Conclusion section should briefly list the conclusions of your research.
4. The limitation section should briefly list the limitations that exist.
Round 2
Reviewer 1 Report
It seems that the paper has gone through several editing processes. After reviewing the article I have found no significant editing issues.
Author Response
Thank you for the opportunity to revise and resubmit our manuscript. Based on your helpful comments and constructive suggestions for improvement, we have made substantial revisions throughout the piece. Sincere thanks!
Reviewer 2 Report
The article is sufficiently improved and it is now suitable for publication. Good job.
Author Response
Thank you for the opportunity to revise and resubmit our manuscript. Based on your helpful comments and constructive suggestions for improvement, we have made substantial revisions throughout the piece. Many thanks!
Reviewer 3 Report
The authors have made changes based on the review comments and I have no further comments about contents.But I still want to remind the author that writing a paper needs to be very serious. There are still many "[Error! Reference source not found.]" in the author's response. This actually shows that the author is not serious about his work.Author Response
Dear reviewer,
Thank you for the opportunity to revise and resubmit our manuscript, “An Empirical Study of Promotion Pressure of University Teachers in China Using Event History Analysis.” Based on your helpful comments and constructive suggestions for improvement, we have made substantial revisions throughout the piece. With your assistance, we are confident we have now improved the overall quality of the article, and we hope you will agree that the work is suitable for publication in IJERPH.
Point 1: The authors have made changes based on the review comments and I have no further comments about contents. But I still want to remind the author that writing a paper needs to be very serious. There are still many "[Error! Reference source not found.]" in the author's response. This actually shows that the author is not serious about his work.
Response 1:
We want to thank you for the encouragement, because that really means a lot to us!!
Concerning the “Error!” problem, we're very sorry for the mistakes,which has brought you inconvenience and unpleasant reading experience.
As a matter of fact, this was no “Error!” when we opened the manuscript, so we guess this may be due to different WORD version problems or other technical problems. We were very serious about our article and worked so hard and tried our best to revise the manuscript as suggested.
So this time, we conducted a comprehensive check thoroughly and carefully, and at the same time uploaded a PDF version as well to avoid no reference errors (see pdf version of the manuscript).
Point 2: English language and style:Moderate English changes required
Response 2:
We would like to thank you for your careful reading and helpful advice. As suggested, we further carefully checked the language several times, and also invited English professional editors and paid for an Urgent Return Surcharge service to help us improve the writing of the manuscript. Please see the invoice attached.
Hope our revisions have appropriately addressed your concerns.
Thanks again for your time and suggestions.
